# The SARS-CoV-2 Pandemic and Cancer Trials Ireland: Impact, Resolution and Legacy [note 1]

**DOI:** 10.3390/cancers14092247

**Published:** 2022-04-30

**Authors:** Seamus O’Reilly, Verena Murphy, Eibhlin Mulroe, Lisa Tucker, Fiona Carragher, Jacinta Marron, Aoife M. Shannon, Ken Rogan, Roisin M. Connolly, Bryan T. Hennessy, Ray S. McDermott

**Affiliations:** 1Cancer Trials Ireland, D02 VN51 Dublin, Ireland; verena.murphy@cancertrials.ie (V.M.); eibhlin.mulroe@cancertrials.ie (E.M.); lisa.tucker@cancertrials.ie (L.T.); fiona.carragher@cancertrials.ie (F.C.); jacinta.marron@cancertrials.ie (J.M.); aoife.shannon@cancertrials.ie (A.M.S.); ken.rogan@cancertrials.ie (K.R.); bryanhennessy74@gmail.com (B.T.H.); ray.mcdermott@tuh.ie (R.S.M.); 2Department of Medical Oncology, Cork University Hospital and Cancer Research @ UCC University College Cork, T12 DCA4 Cork, Ireland; roisin.connolly@ucc.ie

**Keywords:** COVID-19, clinical trials, transformative change

## Abstract

**Simple Summary:**

The SARS-CoV-2 pandemic led to a significant disruption to healthcare. Cancer Trials Ireland is the national cooperative cancer trials organisation in Ireland. We analysed the impact of the pandemic on the conduct of cancer clinical trials over a 2 year period. Clinical trial accrual fell by 54%, with trials in radiotherapy being most affected, declining by 90%. This reduction was due to reduced staffing, delays in trial logistics and halting of accrual and initiations due to safety concerns. Remote monitoring of trials increased. Protocol violations increased due to interruption of visits and diagnostic testing. The impact on accrual persisted for at least 18 months. Significant central office staffing issues were evident with recruitment and retention. A retreat was conducted to identify the transformative changes that were needed to build resilience in clinical trial conduct. Adaptive strategies in trial conduct such as expanded telehealth and tele-monitoring, flexibility of trial scheduling, protected staff assignments, and mentoring of research staff were identified as priorities for the recovery stage of the pandemic.

**Abstract:**

Background: Cancer Trials Ireland (CTI) is the national cooperative group in Ireland. The SARS-CoV-2 pandemic led to significant ongoing disruptive change in healthcare from March 2020 to the present day. Its impact and legacy on a national clinical trials organisation was assessed. Methods: A review was conducted of prospectively acquired communications, team logs and time sheets, trial activation, closure and accrual, for the period 2019 to September 2021. An online survey of the impact of the pandemic on clinical investigators and of clinical trials units was performed. A National Cancer Retreat was organised on 21 May 2021 to identify and address pandemic related disruption and develop adaptive strategies. Results: In the weeks after the pandemic was declared, remote working was initiated by all central office staff. Nationally, clinical trial accrual fell by 54% compared to the same period in 2019, radiotherapy trial accrual by 90%, and translational studies by 36%. Staff reassignment of research nurse staff occurred in 60% of units, trial monitoring was reduced in 42%, and trial initiations fell by 67%. Extreme fluctuations in monitoring hours were noted paralleling lockdown measures. Significant impact on all clinical trials units was noted including staff reassignments, reduced access to diagnostic imaging and reduced institutional supports. Remote clinic visits and remote monitoring was widely adopted. The National Cancer Retreat identified flexibility in trial conduct, staff recruitment and retention, the need for harmonisation of processes, and research staff support in the context of remote working as priorities. Conclusion: The pandemic has had a significant ongoing negative impact on cancer clinical trial activity in Ireland. Adaptive strategies including trial flexibility, expanded telehealth and remote monitoring, harmonisation of processes and staff support have been identified as priorities to ameliorate this impact, and develop a more sustainable clinical trial ecosystem.

## 1. Introduction

The COVID-19 pandemic led to a full national lockdown in Ireland on 27 March 2020, with a gradual release of restrictions in June 2020 [1]. This first wave was followed by two others. The second wave from October to December 2020 and the third from January to May 2021 [2,3,4]. In the second week of January 2021 Ireland recorded the highest weekly rate of infection per 1000 people in the world [2]. COVID-19 resulted in significant healthcare disruption in Ireland as resources were prioritised for COVID-19 related care [4,5,6,7]. Non-COVID-19 related care was impacted through staffing, social distancing measures and nosocomial infection risks associated with healthcare visits [3]. Cancer diagnosis declined nationally by over 10% [8].

The vulnerability of patients with cancer led to efforts to reduce exposure to people with COVID-19 and asymptomatic carriers, while continuing to ensure access to care including clinical trials [9]. COVID-19 had the potential to impact the scientific integrity and patient safety of ongoing trials, increase operational burdens, limit accruals and trial access, particularly for the most vulnerable populations. While six million doses of the COVID-19 vaccine had been administered in Ireland by August 2021, the low rate of vaccination globally, with anticipated global coverage by 2023, indicates that a COVID endemic reality may be present for some time [10].

Cancer Trials Ireland (CTI) (www.cancertrialsireland.ie accessed on 21 April 2022) is Ireland’s national cooperative clinical trials organisation. In this study we analysed the pandemic’s impact on the conduct of cancer clinical trials nationally in order to develop adaptive strategies to facilitate continued trial conduct and accrual, and develop resilience for clinical trial conduct in a COVID endemic reality.

## 2. Methods

CTI has prospectively tracked metrics for cancer trials from 1997 to the present. The metrics from CTI affiliated cancer clinical trials units were analysed for the period from 2019 to 2021 for clinical trial accrual, monitoring activity, trial activations, protocol deviations and COVID-19 related deaths. The life cycle of CTI conducted trials were graphed to assess the temporal impact of COVID-19 related changes on monitoring and administration. CTI associated clinical trials units were surveyed nationally in August 2021 regarding the impact of the pandemic on the conduct of clinical trials using a questionnaire adapted from the American Society of Clinical Oncology [11]. A National Cancer Retreat was conducted on 21 May 2021 to assess the impact of the pandemic on trial conduct and develop strategies to mitigate against COVID-19 impacts [12].

## 3. Results

The ongoing and evolving impact of COVID-19 incidence and National Health Service Executive guidelines directly impacted on clinical trial unit activity (Table 1). 

Staff reassignment was reported in 41% of units surveyed and diagnostic capacity was reduced in 71%. Full or partial accrual suspension was reported in the majority of units. Trial logistics were impacted by delays in ethical approvals and risk management assessments. Research sites had to deal with both the administrative tasks of safe trial conduct, and a reduced clinical workforce due to staff reassignment. Telehealth visiting was initiated accompanied by minimised exposure to healthcare environments through adherence to social distancing guidelines for clinical and treatment visits. Remote monitoring, working and site initiations were widely adopted to reduce the risk of infection to patients and staff. 

The impact of these measures on clinical trial accrual is demonstrated in Figure 1. A 40% reduction in activity was observed in 2020 compared to the previous year. This reduction persisted into 2021. This reduction was multifactorial in aetiology and was related to reduced staffing, halting in trial accrual due to COVID-19 related safety concerns and delays in trial logistics. 

At CTI central office, remote working for all staff was initiated at the onset of the first lockdown. At the start of 2021, 15% of office staff were permitted to return onsite, rising to 40% subsequently. All staff, board, and investigator meetings moved to virtual platforms with the announcement of lockdown, and had not returned to in-person formats by the end of 2021. Mapping exercises of monitoring hours related to individual clinical trial were conducted to explore the impact of COVID-19 and other stressors (e.g., protocol amendments) on central office workload (Figure 2). Monitoring hours were also negatively impacted by Brexit, an event which had implications for risk management, data protection, trial sponsorship and drug distribution pathways.

Staff turnover reduced during 2020, but staff resignations increased in 2021 as the national impact of the pandemic eased. The impact on central office activity is demonstrated in Figure 3. The proportion of remote monitoring hours increased relative to 2019 activity, while overall activity decreased throughout 2020 and continued to fall in the first half of 2021, reflecting reduced clinical trial accrual. 

To assess the impact of these measures on protocol violations and serious adverse events we assessed protocol deviations in studies open to accrual in 2019 and 2020, and compared the number of deviations observed for each period. As shown in Figure 4a, a significant increase in protocol deviations was observed in all studies assessed. The majority of violations related to treatment visits (Figure 4b). Two COVID-19 related deaths were reported in the period to December 2021. Both deaths were in patients with haematologic malignancies.

Recognition of the need for transformative change prompted the first National Cancer Retreat in 2021 to reflect on the challenges posed by the pandemic. This day-long virtual conference featured 30 contributors from Europe and North America across 16 sessions (plenaries, panels, and breakout groups). It was attended by over 250 members of Ireland’s cancer clinical trials community with a view to discuss the cancer trial landscape in Ireland, including choosing trials for Ireland, running, and funding them more efficiently, and a number of recommendations were formulated (Table 2). 

It also presented an opportunity to implement positive change. These changes (Table 3) are multifaceted, ranging from improving operational efficiencies, simplifying regulatory requirements, minimising non-essential tests, to facilitating virtual interactions, such as informed consent and monitoring. 

## 4. Discussion

The COVID-19 pandemic has had a significant and persistent negative impact on the conduct of cancer clinical trials in Ireland. These impacts include reduced initiation of new trials, and accrual, monitoring and conduct of ongoing trials. Change in work practice, with remote working for all central office staff, and staff reassignment of clinical nursing staff in many units made the safe conduct of trials challenging. Telemedicine visits were universally adopted to reduce patient exposure to healthcare environments. These changes were necessary to ensure patient safety. Overall, cancer patients with COVID-19 have a high 30 day all-cause mortality and are 3.5 times more likely to require intensive care unit admission and ventilation [9,13]. Over 24 months since the first national lockdown was initiated, many of these changes persist and are likely to be integrated into future trial conduct.

The experience of the pandemic at CTI was similar to that reported by clinical trial organisations in other jurisdictions. A survey by the American Society of Clinical Oncology demonstrated similar challenges with over 50% of respondents reporting reluctance of patients to attend healthcare environments, and reduced availability of ancillary services, such as radiology, required to conduct trials [11]. A study limited to breast cancer clinical trials in the Unites States reported a 44% reduction in accrual for interventional studies in the first 6 months of the pandemic [14]. In Latin America, a survey of 90 research centres observed that accrual was suspended in at least some studies in 80% of centres, and clinical trial conduct was altered in 96.7% of centres [15]. 

While accrual to CTI conducted clinical trials fell during the pandemic, there remained a duty of care to ensure continued care and treatment of the patients on study at a time of reduced staffing and high clinical vulnerability. Reports from academic cancer centres on the challenges of safely conducting clinical trials during this time [16,17,18] have also emphasised the significant burden on clinical trials units for the safe conduct of trials during the onset of the pandemic. These measures included a four-level restriction escalation plan and a two-stage recovery plan at the Dana Farber Cancer Institute [16] and escalated Protocol Review and Monitoring Committee meetings at the University of Cincinnati [17]. These reports concur with the challenges reported in this paper on the impact of the pandemic on trial logistics, such as patient visits, risk management and monitoring. The significant increase in protocol violations observed in our study compared to previous years reflects the pandemic related challenges of trial provision. Such challenges are particularly relevant given patient safety concerns with COVID-19 infection and the higher mortality rates noted in our patients [6]. 

Pandemics magnify inequalities and expose vulnerabilities. Surveys of Irish healthcare workers during this time have highlighted the disproportionate burden placed on women [19] and non-national doctors in training [20,21]. In Ireland, many nursing staff have hospital contracts as opposed to university contracts, which facilitated clinical reassignment and consequently reduced staffing of clinical trials. COVID-19 associated reassignment of nursing staff occurred in 41% of CTI affiliated units. Resulting staff losses were compensated for by the halting of trial accrual in centres to ensure safe patient care; however, the impact of accrual reduction and the burden on remaining staff may have been less if employment contracts were associated with academic affiliates rather than hospitals. Such disruptive practice will increase burnout and exhaustion for all staff [22,23], compounding the impact of the pandemic. At CTI, central office staff turnover reduced in the initial phases of the pandemic, but as recovery has occurred turnover has now increased. The significant human capital of the organisation reinforced by in-person team building before the pandemic was an important asset during it. However, replacing departing staff and rebuilding a team in a hybrid environment will be challenging [24]. 

In December 2020, vaccination against COVID-19 was initiated in Ireland. The programme has transformed clinical care, with vaccination rates of over 80% nationally [25]; however, recent studies have emphasised that patients with cancer may not respond as readily to vaccination [26] and remain vulnerable, particularly to new viral variants [27]. In our current COVID-19 endemic reality [9], successful clinical trial conduct will involve maintaining flexibility in scheduling and reduced reliance on in-person attendance by patients and monitoring staff. Consequently, many of the measures implemented during the pandemic will endure.

Several groups have emphasised the potential opportunities for long term transformational change to improve patient access to trials based on the experiences of COVID-19 related trial disruption [28,29,30,31,32]. Such change is needed. Clinical trials have traditionally been limited by high costs, lengthy timelines, poor geographical accessibility, significant participant hurdles, and nonrepresentative study populations. The stricter timelines [33] and increasing complexity of trials in recent decades [34] have magnified these challenges. Previously conducted CTI trials, such as TAILORx, and currently enrolling trials, such as DECRESCENDO, are investigating de-escalation of treatment. Reduced geographical access impedes patient opportunity to receive such potentially less intensive treatments, which may improve quality of survival without impacting cancer-free survival. Lower patient accrual for these, alongside other reasons, prolong the time required to determine if new treatments or interventions are effective. This reduces the generalisability of the results and increases the cost of discovery. COVID-19 associated changes such as telemedicine to reduce travel, decentralising study touchpoints to increase patient convenience, and remote informed consent all bring clinical trials nearer to the patient [32,35,36,37]. Trials conducted during the pandemic by the RECOVERY collaborative group [38,39] have demonstrated the success of pragmatic rapidly developed, and completed, clinical trials of treatments for COVID-19. Such trials may serve as templates for future cancer clinical trials, with their associated reduced costs leading to increased access and, consequently, greater generalisability of results. 

Concern that the pandemic would overwhelm healthcare systems [40] led to prompt implementation of widespread rapid changes in Irish healthcare [2,41]. These included universal free health care at the point of delivery for all COVID-19 related diagnoses, and early care and infrastructural improvements. Some changes, such as centralised procurement and introduction of a universal healthcare number, had been advocated for over a decade. The pandemic removed institutional inertia. 

The Cancer Retreat offered a possibility to reflect on the challenges and possible solutions, especially minimising non-essential tests, and facilitating virtual interactions. Doroshow et al. [42] have also emphasised that such changes are imperative if we are to improve clinical trial access for patients. 

## 5. Conclusions

The COVID-19 pandemic led to a significant prolonged reduction in clinical trials accrual by CTI. This reduction was polyfactorial, due to trial suspension, staff reassignments, and delays in trial logistics. The pandemic represents a watershed opportunity to build a more resilient and equitable cancer clinical trials ecosystem nationally. A National Cancer Retreat held in response to the pandemic identified more efficient trial processes, less onerous clinical trial schedules for patients, and mentoring and support systems for clinical trials staff as priorities in this regard. 

As we recover from the pandemic, we will need such resilience in order to deal with the increased burden of delayed diagnosis in our communities, e.g., [8,43] and the challenges of staff recruitment and retention in healthcare [23,44].

## Figures and Tables

**Figure 1 cancers-14-02247-f001:**
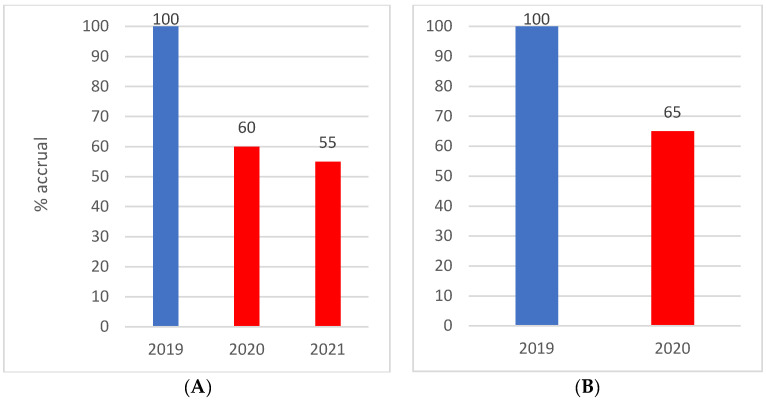
Impact on clinical trial accrual 2019–2021. (**A**) Clinical trial accrual January to June; (**B**) clinical trial accrual January to December.

**Figure 2 cancers-14-02247-f002:**
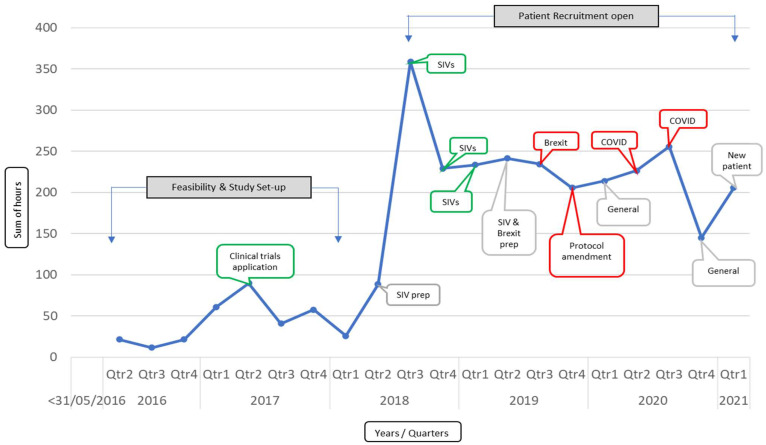
A representative CTRIAL-IE study before and during the pandemic, highlighting the effect of the pandemic (and Brexit) on level of staff resource hours for trial management (x-axis: staff hours, y-axis: years in quarters) Green and grey circled tasks: regular trial related work; red circled tasks: stressors resulting in increased work hours.

**Figure 3 cancers-14-02247-f003:**
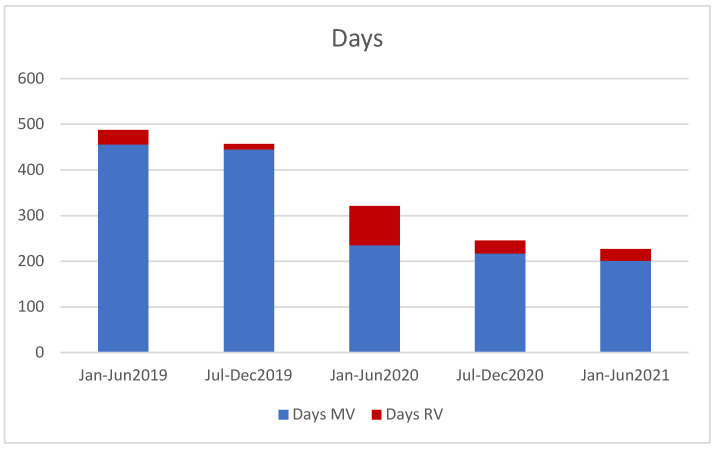
Monitoring activity in person (MV) and remotely (RV) by CTI central office staff, per 6 month period.

**Figure 4 cancers-14-02247-f004:**
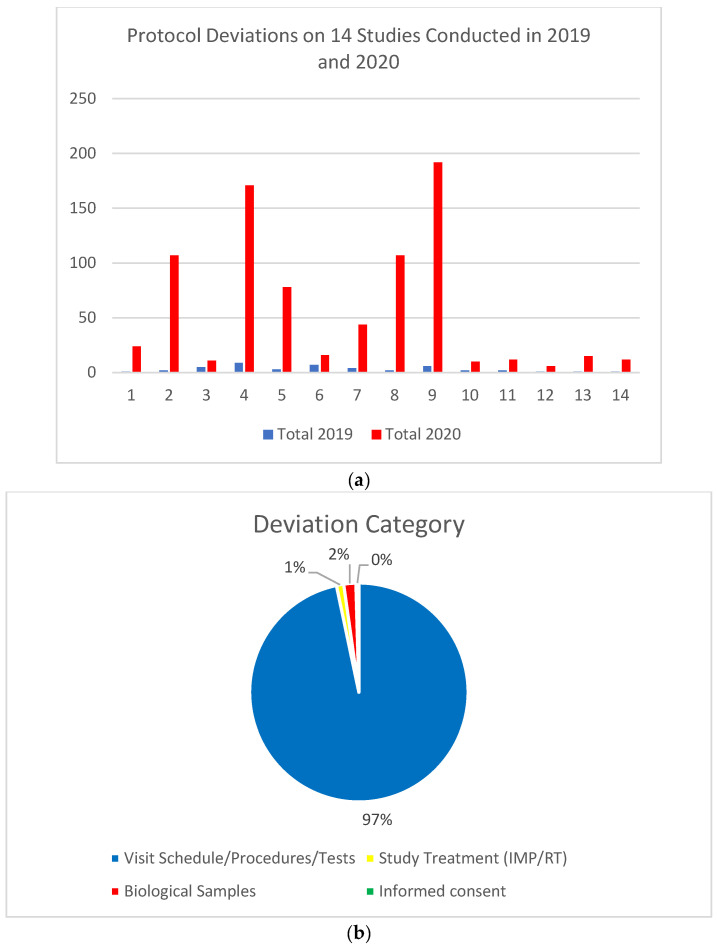
(**a**) Paired assessment of protocol deviations of studies open to accrual in 2019 and 2020; (**b**) categories of protocol deviations observed.

**Table 1 cancers-14-02247-t001:** Impact of the pandemic on the 17 clinical trials units surveyed ^a^.

Activity	Heading	Activity	Heading
Staff reassignment	41%	In person clinic visits reduced	88%
Diagnostics impacted	71%	Telehealth visits increased	71%
Accrual suspended (full)	56%	Onsite monitoring reduced	100%
Accrual suspended (partial)	71%	Physical site initiations of trials reduced	94%
Accrual to trials impacted	81%	Remote monitoring adopted	53%
Research ethics decisions delayed	67%	Remote working adopted	65%
Risk management delayed	31%	Virtual site initiations adopted	44%

^a^ Data reflects % of units where activity was affected.

**Table 2 cancers-14-02247-t002:** Recommendations from the Cancer Retreat.

1. Clinical research must be embedded into wider healthcare planning.
2. Development of a funding stream for translational research.
3. Increase of Public and Patient Involvement (PPI) in cancer trials.
4. Analysis of the future effects of the pandemic on cancer diagnosis.

**Table 3 cancers-14-02247-t003:** COVID-19 associated challenges and solutions to clinical trial conduct.

Challenges	Solutions
Reassignment of research staff	Research unit specific contracts of employment, e.g., involvement in clinical trials is part of the contract
Limited networking opportunities	Regular topic specific virtual meetings with in-person meetings when public health guideline acceptable
Recruitment and retention in a virtual workplace	Assigned mentorship by organisation leadership
Homeworking for clinical trial staff	Cyber secure clinical trials information technology access, e.g., providing IT equipment, which can be used securely outside the hospital
In-person hospital visits for trial assessments and product delivery	Integration of telehealth visits into protocols in-lieu of attendance with home delivery of trial products
Trial schedule disruption due to COVID-19 related absences	Flexibility of trial assessments without hampering treatment safety
Burnout and exhaustion among investigators	Health and wellbeing support, reassignment of tasks

## Data Availability

The data cited in this report are retained by Cancer Trials Ireland.

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
