# Peer review of "The SARS-CoV-2 Pandemic and Cancer Trials Ireland: Impact, Resolution and Legacy†"

_cancers, 2022, doi:10.3390/cancers14092247_

Round 1

Reviewer 1 Report

O'Reilly et al report entresting resultsI suggest to shorten the discussion section

Author Response

please see attachment - discussion shortened as requested

additional points raised by reviewer 2 included

Reviewer 2 Report

I reviewed the manuscript "The SARS Cov2 Pandemic and Cancer Trials Ireland: Impact, Resolution, Legacy." The authors do the fields dependent upon clinical trials are huge service through this undertaking. I have some high-level organization critiques that should help improve the readability and usefulness of this paper.

In the Simple Summary and Abstract, the authors highlight a retreat that was held to identify transformative changes that were needed to build resilience in clinical trial conduct. I found it strange that the recommendations from this retreat were not included in the results section and only discussed briefly in the discussion. I encourage the authors to present this paper as a two-pronged tool for individuals around the globe who are involved in clinical trial design and implementation. The authors have a helpful summary of how and how many clinical trial units were impacted by the pandemic. They provide details on the experiences of the staff members and how that influenced clinical trial-related activities. I'm wondering if they can pair those challenges with the findings from the retreat? So a summary of challenges experienced followed with solutions provided by those who experienced those challenges most acutely.

Related to the point above, I'm also wondering if the authors can provide more specific examples of the recommendations made in Table 2? I can see how those responsible for implementing clinical trials would want some clear direction from these findings so increased precision here will be very useful. For the first solution noted, what specifics of a contract should be investigated? What can be done when those specifics are identified? And I have the same advice for the remaining recommendations.

I would also like to see more detail on the retreat, who was involved, and how recommendations were determined?

Some further orientation to Figure 2 would also be helpful. I understand what is happening on the x-axis but what does the y-axis represent? Is that resource intensiveness? And for those readers who are less familiar with Brexit and its impact on Ireland, I am hopeful the authors could provide a brief summary of Brexit and why specifically it should be considered while examining clinical trials.

Author Response

thank you for your helpful comments which we have used to improve the manuscript - attached below are details - we appreciate the time and interest invested in the review

I reviewed the manuscript "The SARS Cov2 Pandemic and Cancer Trials Ireland: Impact, Resolution, Legacy." The authors do the fields dependent upon clinical trials are huge service through this undertaking. I have some high-level organization critiques that should help improve the readability and usefulness of this paper.

In the Simple Summary and Abstract, the authors highlight a retreat that was held to identify transformative changes that were needed to build resilience in clinical trial conduct. I found it strange that the recommendations from this retreat were not included in the results section and only discussed briefly in the discussion.

New table (2) added and discussion of the retreat embedded into the methods and results section of the document

I encourage the authors to present this paper as a two-pronged tool for individuals around the globe who are involved in clinical trial design and implementation. The authors have a helpful summary of how and how many clinical trial units were impacted by the pandemic. They provide details on the experiences of the staff members and how that influenced clinical trial-related activities. I'm wondering if they can pair those challenges with the findings from the retreat? So a summary of challenges experienced followed with solutions provided by those who experienced those challenges most acutely.

This table was already in the discussion session. We have Moved this and associated  text in the results section. This also covers comments from reviewer no 1 requesting shortening of the discussion .

Related to the point above, I'm also wondering if the authors can provide more specific examples of the recommendations made in Table 2? I can see how those responsible for implementing clinical trials would want some clear direction from these findings so increased precision here will be very useful. For the first solution noted, what specifics of a contract should be investigated? What can be done when those specifics are identified? And I have the same advice for the remaining recommendations.

We have added additional wording to highlight opportunities such as harmonisation of trial processes and remote access for patients – these are similar recommendations to those highlighted by other groups both clinical trail sites and cooperative groups.

I would also like to see more detail on the retreat, who was involved, and how recommendations were determined?

We have added to the text about the retreat in the result section, reference to retreat report hyperlinked in reference section

Some further orientation to Figure 2 would also be helpful. I understand what is happening on the x-axis but what does the y-axis represent? Is that resource intensiveness? And for those readers who are less familiar with Brexit and its impact on Ireland, I am hopeful the authors could provide a brief summary of Brexit and why specifically it should be considered while examining clinical trials.

Figure re-done and old one replaced with this with axis orientation which reflects hours of monitoring required to manage changes. A discussion on the impact of brexit is added to the text – brexit impacts arose as we are in the European Union and remain so – BREXIT resulted in changes in drug distribution pathways, risk management and data protection.

As requested by the journal all references have been reviewed and amended to include first 10 authors

Round 2

Reviewer 2 Report

Thank you for revising based on the previous critiques and suggestions.